# Guanylation Reactions for the Rational Design of Cancer Therapeutic Agents

**DOI:** 10.3390/ijms241813820

**Published:** 2023-09-07

**Authors:** Almudena del Campo-Balguerías, Blanca Parra-Cadenas, Cristina Nieto-Jimenez, Iván Bravo, Consuelo Ripoll, Elisa Poyatos-Racionero, Pawel Gancarski, Fernando Carrillo-Hermosilla, Carlos Alonso-Moreno, Alberto Ocaña

**Affiliations:** 1Unidad nanoDrug, Centro Regional de Investigaciones Biomédicas, Universidad de Castilla-La Mancha, 02008 Albacete, Spain; almudena.delcampo@uclm.es (A.d.C.-B.); ivan.bravo@uclm.es (I.B.); consuelo.ripoll@uclm.es (C.R.); 2Departamento Química Inorgánica, Orgánica y Bioquímica, Facultad de Farmacia de Albacete-Centro de Innovación en Química Avanzada (ORFEO-CINQA), Universidad de Castilla-La Mancha, 02008 Albacete, Spain; 3Departamento de Química Inorgánica, Orgánica y Bioquímica-Centro de Innovación en Química Avanzada (ORFEO-CINQA), Universidad de Castilla-La Mancha, 13071 Ciudad Real, Spain; blanca.parra@uclm.es (B.P.-C.); fernando.carrillo@uclm.es (F.C.-H.); 4Experimental Therapeutics Unit, Hospital Clínico San Carlos, IdISSC, Fundación Jiménez Díaz, START, 28040 Madrid, Spain; 5Departamento Química-Física, Facultad de Farmacia de Albacete, Universidad de Castilla-La Mancha, 02008 Albacete, Spain; 6Cancerappy, Avda Ribera De Axpe, 28, 48950 Erandio, Spain; elisa.poyatos@cancerappy.com (E.P.-R.); pgancarski@cancerappy.com (P.G.)

**Keywords:** breast cancer, guanidines, guanylation reactions, high-throughput testing

## Abstract

The modular synthesis of the guanidine core by guanylation reactions using commercially available ZnEt_2_ as a catalyst has been exploited as a tool for the rapid development of antitumoral guanidine candidates. Therefore, a series of phenyl-guanidines were straightforwardly obtained in very high yields. From the in vitro assessment of the antitumoral activity of such structurally diverse guanidines, the guanidine termed *ACB3* has been identified as the lead compound of the series. Several biological assays, an estimation of AMDE values, and an uptake study using Fluorescence Lifetime Imaging Microscopy were conducted to gain insight into the mechanism of action. Cell death apoptosis, induction of cell cycle arrest, and reduction in cell adhesion and colony formation have been demonstrated for the lead compound in the series. In this work, and as a proof of concept, we discuss the potential of the catalytic guanylation reactions for high-throughput testing and the rational design of guanidine-based cancer therapeutic agents.

## 1. Introduction

Guanidines are attractive motifs in medicinal chemistry [1]. For drug designers, their easy-tunable physicochemical properties, protonatability in biological media, and capacity to bind very different targets provide a valuable core to be explored for high-throughput testing in the development of therapeutic agents [2,3]. In this regard, drugs containing guanidine moieties already constitute remarkable pharmaceuticals: rosuvastatin, an anti-inflammatory drug to prevent cardiovascular disease [4]; metformin, an antidiabetic drug [5]; guanabenz, an antihypertensive drug [6]; cimetidine, a drug clinically used to treat peptic ulcers [7]; methotrexate, an antirheumatic drug for rheumatoid arthritis [8]; and antimicrobial drugs such as the antibiotics streptomycin [9], the antiviral drug zanamivir, effective against influenza [10], and the antimalarial drug proguanil [11] (see molecular structures in Figure 1). 

In cancer research, some drugs containing the guanidine moiety have been approved for clinical use, including the tyrosine kinase inhibitors such as Imatinib, to treat chronic myelogenous leukemia and gastrointestinal stromal tumors [12]; Pazopanib, for renal cell carcinoma and soft tissue sarcoma [13]; Nilotinib, approved to treat Philadelphia chromosome-positive chronic myelogenous leukemia [13]; and Gefitinib [14], for the treatment of non-small cell lung cancers. Other guanidines are still in preclinical studies: MIBG and MGBG as mitochondrial function inhibitors [15,16], the antitumoral phenyl-guanidine derivatives for the inhibition of *Rac1* [17], other intercalative drugs [18], and minor groove binders [19,20,21,22]. 

The conventional methods of obtaining guanidines rely on classical organic synthesis [23]. These methodologies are based on chemical transformations of electrophilic reagents, such as thioureas, isothioureas, carbodiimides and cyanamides, pyrazole-1-carbodiimides, sulfonic acids, benzotriazoles, and activated imidazole compounds (Figure 2). However, drawbacks such as the use of toxic solvents, costly reagents, poor availability of precursors, low yields, and production of undesirable substances highlight the need for more efficient syntheses. Even though the use of derivatives of pyrazole-1-carboximidamide, and isothiourea and thiourea derivatives, is the most common methodology, the introduction of a protecting group is required, which brings about synthetic difficulties associated with the optimization of the reaction conditions and purification of the desired guanidines. Likewise, the reaction of protected thioureas with primary and secondary amines demands the presence of a metal salt to accelerate the desulfurization. Thiourea-derived sulfonic acids and carbodiimides for chemical transformation into guanidines currently demand improvements in yields and one-pot procedures. The use of pyrazole carboximidamide transfer reagents results in poor yields along with the requirement of multiple synthetic steps with costly reagents, and the sequential displacement of benzotriazole or imidazole by amines gives rise to moderate yields. Due to the acute toxicity of cyanogen bromide, the use of cyanamide precursors is unlikeable. Finally, the use of triflyl guanidine reagents is limited by the availability of the starting guanidines. 

In this context, the catalytic guanylation of amines with carbodiimides provides an efficient and atom-economical way for the synthesis of these entities (Figure 3) [24]. There are numerous metal-based catalysts for the generation of guanidines [24,25]. Among them, the use of commercially available products for guanylation reactions, such as ZnEt_2_ [26], might ease extending the scope of guanidine-based molecules. 

Cancer is one of the main causes of death worldwide and, particularly in solid tumors, constitutes the most relevant group, as limited therapeutic options exist in the advanced setting. With the incorporation of novel chemotherapies, targeted agents, and immunomodulators, the outcome of cancer patients has improved in specific indications. However, for some solid tumors, such as triple-negative breast cancer, small-cell lung cancer, head and neck cancer and bladder cancer, there are unmet needs in their metastatic settings. In this context, even though some compounds seem promising due to their antitumoral efficacy profile, relevant toxicity limits their clinical development [27]. 

Classical treatments for cancer include chemotherapy, radiotherapy, and targeted agents. However, many patients harbor an inherent resistance to drugs [28,29], and progression becomes frequent over time. In this regard, the identification of new agents with higher efficacy and a safer toxicity profile is needed, and tools for the straightforward synthesis of molecules with potential antitumoral activity are worth studying. To date, phenyl-guanidine derivatives have been provided for the inhibition of Rac1 as agents for the treatment of aggressive and resistant tumors [17]. The synthesis of such entities, based on classical organic synthesis, requires cyanamides as source materials and intense purification to give rise to moderate yields. Some of us have reported a family of phenyl-guanidines obtained from catalytic guanylation reactions as a potential therapy for glioblastoma [30]. During testing, a lead compound (*ACB0*) was identified with IC_50_ values 16 times lower than temozolomide, the reference compound used for the treatment of glioblastoma (Figure 4). 

Herein, and to enhance the antitumoral activity of that lead compound, modifications of the structure were accomplished through guanylation reactions utilizing a catalyst and starting reagents commercially available. The synthesis using guanylation reactions was straightforward, allowing us to obtain a family of potential antitumoral agents. The compounds were evaluated against a panel of tumoral cells to identify the new lead compound. Biochemical studies and fluorescence time-resolved microscopy were carried out to explore the mechanism of action, and ADME parameters were estimated to propose a structure–activity relationship. This work might set the basis for further improvements in guanidine-based drugs for cancer research.

## 2. Results

### 2.1. Synthesis of Guanidine-Based Agents

Main group metal compounds available are a remarkable group of catalysts for guanylation reactions. Since the pioneering work of Richeson et al., in which lithium hexamethyldisilazide, (LiN(SiMe_3_)_2_), was used as a catalyst precursor for the guanylation of aromatic amines with carbodiimides [31], other simple and commercially available catalysts have shown excellent yields even at room temperature [24]. In this sense, guanidines *ACB0-ACB10* were obtained by guanylation reactions using the cheap and commercially available ZnEt_2_ as a catalyst. The synthesis involved the formation of a zinc alkyl-amido species, which is formed by the insertion of a carbodiimide molecule into the Zn–N bond of an amido complex that, in turn, is formed by the reaction between ZnEt_2_ and the amine substrate, followed by posterior nucleophilic addition to the carbodiimide and amine protonolysis of the guanydinate species. At the outset, guanidines *ACB1-ACB10* were obtained to expand the previously covered structure space [30], with the aim of generating new guanidine-core therapeutic agents (Figure 5). Due to the abundance of commercial amines and carbodiimides, this methodology is very versatile in expanding guanidine sets for cancer screening. The guanidines *ACB0–ACB10* were obtained in very high good yields after 2 h of reaction. In the first step, the catalyst is activated by the addition of the corresponding amine at a mild temperature to give rise to the amide derivative. Later, *N*,*N*′-dicyclohexylcarbodiimide was added to the above reaction mixture to provide the solid guanidine products. Guanidines *ACB0-ACB6* and *ACB8-ACB9* were previously reported [24]. The characterization of the novel guanidines *ACB7* and *ACB10* was carried out by analytical methods, infrared (IR), mass spectroscopy and nuclear magnetic resonance (NMR) spectroscopy. Structural elucidation is depicted in the experimental section and illustrated in Appendix A.

### 2.2. Antiproliferative Activity of Guanidines ACB0-ACB9 in Cancer Cell Lines

We evaluated the antitumoral activity of guanidines *ACB0-ACB9* in three different cellular models, including MDA-MB-231, a classical model of triple-negative breast cancer, OVCAR-8, an ovarian cancer model, and SW-620, a colorectal cancer cell line. As shown in Figure 6A–C, some compounds needed high concentrations to induce 50% of cell death with doses above 10 µM. On the other hand, the compound *ACB3* demonstrated remarkable activity inducing 50% of cell death in more than half of the cell population with doses around 4 µM (Figure 6D). These data demonstrate that *ACB3* has more effect and can be selected for further evaluation. In all cases, the incorporation of cyclohexyl groups into the guanidine core improved the cytotoxicity of the previously identified lead compound: *ACB0* (see antitumoral activity in Appendix A).

### 2.3. ACB3 Induces Cell Cycle Arrest and Reduces Cell Adhesion and Survival

*ACB3* was selected for further evaluation of the mechanism of action given its promising antitumoral activity. In the first step, we evaluated the effect of this guanidine on the cell cycle, by treating MDA-MB-231 and SW-620 with *ACB3* at 10 μM for 24 h. As shown in Figure 7A, *ACB3* was able to induce cell cycle arrest at G0/G1 in all three evaluated cell lines.

*ACB3* was also able to demonstrate a reduction in cell adhesion in MDA-MB-231 and OVCAR-8 (Figure 7B). This effect could not be evaluated in SW-620, as this cell line does not adhere to the petri dish [32]. Finally, the action of the compound on long-term viability was evaluated with the help of clonogenic assays. Treatment with *ACB3* reduced the number of colonies in a statistically significant manner in both MDA-MB-231 and OVCAR-8 (Figure 7C).

### 2.4. Treatment with ACB3 Leads to a Strong Induction of Apoptosis

Next, we explored the effect of *ACB3* on the induction of cell death. To do so, we treated MDA-MB-231, OVCAR-8, and SW-620 at 10 μM for 72 h, observing a profound induction of apoptosis in all three evaluated cell lines (Figure 8A–C for MDA-MB-231, OVCAR-8, and SW-620, respectively). The major effect was identified with MDA-MB-231, where the activity was mainly mediated by the induction of early apoptosis (Figure 8A).

### 2.5. ADME Analysis for Providing Activity–Structure Relationships

In silico prediction of absorption, distribution, metabolism, and excretion (ADME) properties was estimated for *ACB1-ACB9* to support the high-throughput screening of guanidines and to provide insights into the structure–activity relations of the obtained guanidines (see Appendix A). Two correlation coefficients were used to estimate how well the different ADME properties correlate with the obtained cell viability (Figure 9A). Pearson and Spearman coefficients are typically used for feature selection/dimensionality reduction in datasets with small n. Pearson tends to identify linear patterns, while Spearman tests for the monotonicity of the relation. Both methods return values in the range of [−1, 1], where 0 is a random noise, while −1 and 1 represent a perfect negative or positive pattern. Pearson, being more restrictive, tends to be more resistant to false positives, especially in low n datasets. The data were analyzed using the Python scipy.stats.

Figure 9A shows very strong (absolute values > 0.7) correlations for parameters related to solubility and size of the molecules. The negative correlation with *log P* means that the higher log *p* values are correlated with the reduction in cell viability (Figure 9B). This could be the result of the compounds’ better ability to penetrate cellular membranes or a property specific to this family of compounds where their affinity happens to be related to lipophilia.

### 2.6. Uptake Studies for a Luminescent Guanidine Derivative

*ACB10* was synthesized by guanylation reaction using ZnEt_2,_ as performed before for *ACB0-ACB9*. The novel derivative was fully characterized by IR and NMR spectroscopy and elemental analysis (see experimental section and Appendix A). Its antitumor activity was assessed in MDA-MB231 cell lines, and its IC_50_ value was also measured (see experimental section and Appendix A).

The luminescent properties of *ACB10* allow us to monitor the cellular uptake of such entities, as a proof of concept, by fluorescence microscopy to gain insight into their mechanism of action. To do so, fluorescence lifetime imaging microscopy (FLIM) was used to monitor the population distribution of *ACB10* in human breast cancer MDA-MB231 cell lines (Figure 10A). The use of FLIM microscopy presents advantages over conventional fluorescence [33]. For instance, the high sensitivity of the fluorescence lifetime to both the intracellular environment and possible changes in the chemical structure of the fluorescent molecule makes it possible to easily monitor chemical transformations that the drug or prodrug may undergo within each compartment and/or cellular organelle. The FLIM images collected in Figure 8A show that *ACB10* is easily internalized in the cell. Thus, at 1 h, it is found and distributed homogeneously within the cytoplasm, with a narrow overall average lifetime distribution spanning 9–19 ns and centered at ~13.5 ns and mapped to color green (Figure 10B). At longer incubation times, no changes in the overall distribution of lifetimes in the cytoplasm are observed suggesting no chemical modification. Although several aggregates of one micron are formed in the cytoplasm, probably related to accumulation in lipid droplets, endosome formation, or interactions with the endoplasmic reticulum. *ACB10* is not observed in the cell nucleus at any time. At 48 h, cell damage is observed and the overall average lifetime distribution is shifted to lower lifetime values (color blue). These findings, together with the previously described results, suggest that the mechanism of action of the guanidine derivatives studied herein is not at the cell nucleus level, as suggested by other works on guanidine derivatives [18,20], but rather is probably more complex and related to inhibitory pathways at the level of the endoplasmic reticulum.

## 3. Discussion

Cancer is a global problem due to the high prevalence and limited therapeutic curative options in most clinical situations. Several indications and clinical scenarios are considered unmet needs due to the lack of therapies; among them, advanced ovarian cancer, breast cancer, and colorectal tumors can be included in this category. Therefore, the improvement of existing therapies and the generation of novel ones are mandatory. To tackle this challenge, screening compound libraries still supports drug discovery. In the field of antitumoral agents, the rapid development of pharmaceuticals requires high-throughput synthesis and rational design. To do so, easy synthetic approaches for the generation of novel structures are mandatory.

Guanidine structures have been the cornerstone of many pharmaceuticals [1]. The devotion to such structures has allowed drug designers to find successful therapies to prevent cardiovascular diseases, treat diabetes, hypertense, peptic ulcers, and even rheumatoid arthritis [2,9,24]. In cancer, the structure of many drugs relies on a guanidine core, such as Imatinib [12], Gefitinib [14], or Nilotinib [13]. The classical synthesis of guanidines is based on approaches that require toxic, poorly available, and costly reagents to give rise to low yields for a comprehensive array of substrates and produce undesirable substances [24,25]. In this regard, guanylation reactions might offer efficient alternatives for synthesizing these systems. Therefore, phenyl-guanidine derivatives *ACB1-ACB9* were obtained in excellent yields using a catalytic guanylation reaction with 100% atom economy in a waste-free process from relatively cheap and widely available starting materials. The design of these small molecules is supported by a first screening of guanidines reported for our research group, in which a lead compound was identified (the guanidine named *ACB0* in this work) [30]. In our previous work, we observed that a phenyl ring in the guanidine core was essential to display significant antitumoral activity, as demonstrated by Lorenzano Menna et al. [17]. In our first work, the commercially available N,N′-Diisopropylcarbodiimide was used as a scaffold for the generation of guanidine derivatives. Since a general trend was observed for those derivatives in which a high level of cytotoxicity correlated with higher lipophilicity values, the commercially available N,N′-Dicyclohexylcarbodiimide was selected for the design of the entities in this study. Numerous factors could be involved in the antitumor activity of such compounds, not only in terms of the physical and chemical properties of the drugs but also their affinity for the drug target or any plausible interaction with other biomacromolecules. However, we hypothesized that the higher lipophilicity of the cyclohexyl rings on the guanidine core might be a plausible starting point to work with and aim to improve the pharmacological profile of the proposed structures. Since we previously identified a lead mono-substituted guanidine-derivate compound, in which the substitution on the phenyl ring took place in the *para* position, in this work, other guanidines (*ACB1-ACB10*) with the same substitution pattern in the phenyl ring but with cyclohexyl groups were proposed.

*ACB1-ACB9* were tested in several tumor cell lines with the aim of identifying the lead compound. *ACB3* proved to be among the most active guanidine of the series. The structure of the lead compound identified, *ACB3*, correlates with the structure of the previous lead compound with a tertbutyl group in the *para* position of the phenyl ring (*ACB0* guanidine). *ACB3* showed more than 4 times the cytotoxicity displayed by the isopropyl-guanidine lead compound *ACB0* against the different cell lines used in the current survey (see Appendix A). Indeed, our results suggest that slight modifications in the guanidine core are decisive for tuning antitumoral activity. An interesting finding was that *ACB3* displayed considerable antitumoral efficacy in ovarian cancer, TNBC, and colon cancer, indicating that these agents could address a wide spectrum of therapeutic needs. Biological assays showed that the mechanism of action was mainly mediated by apoptotic cell death and an increase in G0/G1 was detected, supporting *ACB3* as a cytotoxic rather than a cytostatic entity. *ACB3* was also able to reduce cell adhesion in MDA-MB-231 and OVCAR-8 and in the number of colonies, indicative of a long-term effect. These data confirm the antitumoral activity of *ACB3* in three different cell line models describing the profound induction of apoptosis.

To gain insight into the mechanism of action of these guanidine molecules, a fluorescent guanidine *ACB10* was designed and obtained by guanylation reaction. *ACB10* displayed equipotent cytotoxicity than the lead compound of the series (see Appendix A). Such entity was used as a dye to monitor the cellular uptake and population distribution in MDA-MB231 cell lines, as a proof of concept, by Fluorescence lifetime imaging confocal microscopy (FLIM). FLIM results show that *ACB10* reaches the cytoplasm easily at 1 h and remains there after 24 h of treatment, showing cell damage at 48 h. Finally, there are novel options to improve drug discovery associated with the current capabilities of integrating massive information using bioinformatic tools and big data. Recently, the rapid growth of computational tools for drug discovery, including anticancer therapies, has exhibited a significant and outstanding impact on anticancer drug design and has also provided fruitful insights into the area of cancer therapy [34]. In this work, in silico tools were used for the estimation of the ADME parameters of the candidates. A correlation between the experimental cytotoxicity obtained values and lipophilia was found. This finding, based on the chemical structure of active small molecules assisted by in silico tools, might pave the way for the rational therapeutic agent design using the guanidine core.

## 4. Materials and Methods

### 4.1. General Procedure

Synthesis reactions were performed using standard Schlenk and glove-box techniques under an atmosphere of dry nitrogen. After using the solvents, they were degassed and distilled from appropriate drying agents. CDCl_3_ was stored over activated 4 Å molecular sieves and degassed by several freeze-thaw cycles. All NMR experiments were conducted in deuterated solvents at 297 K in a Varian FT-400 spectrometer (VARIAN Inc., Palo Alto City, CA, USA) equipped with a 4 nucleus ASW PFG ^1^H/^19^F/^13^C/{^15^N-^31^P}. The ^1^H π/2 pulse length was adjusted for each sample. ^1^H- and ^13^C{^1^H}-NMR chemical shifts (δ) are given in ppm relative to TMS. Coupling constants (*J*) are documented in Hz. The solvent signals were used as references and chemical shifts were converted to the TMS scale. IR experiments were conducted on an FT/IR-4000 Series Jasco Instruments (Jasco Analytics, Madrid, Spain). The UV-Vis absorption spectra were recorded at room temperature using a Cary 100 spectrophotometer (Agilent, Madrid, Spain) using a slit width of 0.4 nm and a scan rate of 600 nm/min. Microanalyses were performed with a PerkinElmer 2400 CHN analyzer.

### 4.2. Synthesis of Guanidine Derivatives ACB1-ACB9

ACB0-ACB10 compounds were obtained following the procedure previously reported [30]. Briefly, 0.03 mmol of ZnEt_2_ (1 M in hexanes) was added over a solution in THF containing 2.00 mmol of the amine under continuous stirring. After 1 h of reaction, 2 mmol of N,N′-dicyclohexylcarbodiimide was added to the reaction mixture and left at 50 °C for 2 h. Then, the solvent was removed under a vacuum and the product was washed with hexane. ACB0-ACB10 compounds were obtained in very high yields as powered yellowish solids. ACB0-ACB6 and ACB8-ACB9 were previously reported [24].

Structural characterization of 1,3-dicyclohexyl-2-(4-(dimethylamino)phenyl)guanidine (ACB7). Yield: 130 mg, 0.38 mmol, 95%. ^1^H NMR (400 MHz, CDCl_3_): δ 6.76 (m, 4H, CH-Ar), 3.61, (bs, 2H, NH), 3.40 (bs, 2H, CH-cyclohexyl), 2.87 (s, 6H, CH_3_-N-Ar), 2.00, 1.69, 1.66, 1.33, 1.10 (5 m, 20 H, CH_2_-cyclohexyl). ^13^C{^1^H}-NMR (101 MHz, CDCl_3_): δ 150.8 (1C, quaternary-guanidine core), 146.3, 110.1 (1C each, quaternary carbons aromatic ring), 124.2 (2C, CH-Ar), 114.8 (2C, CH-Ar), (1C, quaternary carbon), 50.3 (2C, CH-cyclohexyl), 41.7 (2C, CH_3_-NH-Ar), 34.0, 25.8, 25.1 (10C, CH_2_-cyclohexyl). UV-vis: maximum absorbance at 268.40 nm. IR: 2919 cm^−1^ (C-H sp^2^ stretching), 2792 cm^−1^ (C-H sp^3^ stretching), 1614 cm^−1^ (C=N stretching), 1506–1442 cm^−1^ (two bands C=C aromatic stretching), 1269–1163 cm^−1^ (two bands C-N aril stretching). MS (ESI) (m/z): 343.2846 (M+ H+, 100%). Elemental analysis calculated (%) for C_21_H_34_N_4_: C, 73.64; H, 10.01; N, 16.36; found C, 73.70; H, 10.22; N, 10.48.

Structural characterization of 2-(anthracene-2-yl)-1,3-dicyclohexylguanidine (ACB10). Yield: 364.5 mg, 0.91 mmol, 50%. ^1^H NMR (400 MHz, CDCl_3_): δ 8.32, 8.20 (s, H_10_, H_7_), 7.95 (m, 3H, CH-anthracene), 7.35 (m, 3H, CH-anthracene), 7.12 (dd, J_HH_ = 8.9, 2.0 Hz, H_13_), 3.79 (bs, 2H, NH), 3.48 (bs, 2H, CH-cyclohexyl), 2.05, 1.69, 1.60, 1.35, 1.14 (5m, 20H, CH_2_-cyclohexyl). ^13^C{^1^H}-NMR (101 MHz, CDCl_3_): δ 150.4 (1C, quaternary-guanidine core), (1C, quaternary-1-anthracene), 147.7, 133.8, 132.00, 130.5 (1C, quaternary-anthracene), 129.3 (1C, *C*H-anthracene), 129.0 (1C, quaternary-anthracene), 128.3, 127.8, 126.6, 126.0, 125.3, 124.3, 124.0, 117.6 (1C, *C*H-anthracene), 50.4 (2C, *C*H-cyclohexyl), 34.00 (4C, *C*H_2_-cyclohexyl), 25.78 (2C, *C*H_2_-cyclohexyl), 25.06 (4C, *C*H_2_-cyclohexyl). UV-vis: maximum absorbance at 257.96 nm, 342.27 nm, 359.8 nm, and 379.45 nm. IR: 2921 cm^−1^ (C-H sp^2^ stretching), 1606 cm^−1^ (C=N stretching), 1487–1446 cm^−1^ (two bands C=C aromatic stretching), 1306–1164 cm^−1^ (two bands C-N aril stretching). Elemental analysis calculated (%) for C_27_H_33_N_31_: C, 81.16; H, 8.32; N, 10.52; found C, 81.51; H, 8.21; N, 10.99.

### 4.3. Biological Assays

Cell lines culture and drugs. Triple-negative breast cancer and ovarian cell lines, MDA-MB-231 and OVCAR-8, respectively, were cultured in DMEM, whereas colon cancer cell line SW-620 was cultured in RPMI medium, both supplemented with 10% inactivated fetal bovine serum (FBS) (Gibco, Waltham, MA, USA) and antibiotics (100 U/mL penicillin and 100 μg/mL streptomycin) (Sigma-Aldrich, St. Louis, MO, USA). The cell lines were maintained at 37 °C in a 5% CO_2_ atmosphere. Cells authenticity was confirmed by STR analysis. The drugs used for the cell treatments were the previously synthesized guanidines. These molecules were dissolved in dimethyl sulfoxide (DMSO) to a concentration of 10 mM each and kept in the −80 °C chamber.

Cell proliferation, adhesion, and colony-formation experiments. Cells were seeded in 48-well plates (10,000 cells per dish) and were treated 24 h later with different doses. After 72 h of treatment, the cell proliferation assay was evaluated through MTT (3-(4,5-dimethylthiazol-2-yl)-2,5 diphenyltetrazolium bromide) (Sigma Aldrich). Cell medium was replaced with red phenol-free DMEM containing MTT (0.5 mg/mL) and incubated for 45 min at 37 °C. DMSO was then added to solubilize the samples, removing the MTT solution beforehand. Absorbance values were recorded in a multi-well plate reader (AMR-100, Allsheng) (A570 nm).

To assess the effect of treatment on colony formation, 150,000 cells were cultured per dish. The following day, cells were treated for 24 h and were then collected, counted, and reseeded in triplicates (500 cells per well) for each condition. After 9 days, cells were stained with crystal violet (0.05%, 10 min). Colonies were quantified using Image J software (1.8.0 version number, National Institutes of Health and the Laboratory for Optical and Computational Instrumentation (LOCI, University of Wisconsin)).

For the adhesion assay, 150,000 cells were cultured on 6-well plates and were treated 24 h later. Simultaneously, fibronectin (10 ug/mL, PBS) was added to 24-well plates for 1 h at 37 °C. Fibronectin was then removed and plates were blocked with PBS + 0.5% BSA for 45 min. Finally, the plates were washed (PBS + 1% BSA) and exposed to a cold shock. After 24 h of treatment, cells were trypsinized, counted, and reseeded in fibronectin-coated plates (100,000 cells per dish). The cells were incubated for 30 min at 37 °C. The medium was then removed, and adhered cells were washed three times (PBS + 0.5% BSA) and stained with crystal violet (0.05%, 10 min on orbital shaker). To solubilize the samples, 10% acetic acid was used, and absorbance was measured at A570 nm using a spectrophotometer multi-well plate reader (AMR-100, Allsheng).

Flow cytometry experiments. To determine the effect on the cell cycle, 250,000 cells were cultured per well. The following day, cells were treated for 24 h and were then washed with PBS by a previous centrifugation. The samples were fixed with 70% ethanol in PBS (30 min, 4 °C). Cell pellets were washed in PBS + 2% BSA and stained each with 300 ul of the staining solution in PBS containing 7-amino-actinomycin D (7AAD) (5 μL per sample) and RNAse (0.05 μg/mL) (1 h, 4 °C, in the dark).

For cell death and apoptosis studies, cells were seeded at a density of 250,000 cells per dish. After 72 h of treatment, adherent and floating cells were trypsinized and collected. Following a wash with PBS, each condition was stained with 300 ul of Annexin Binding Buffer containing Annexin V-DT-634 (5 ul per each sample), 7AAD (5 ul per each sample), and RNAse (0.05 ug/mL) (1 h, in darkness).

All the samples were analyzed by a CytoFLEX flow cytometer using the CytExpert software (2.3.0.84 version number. Beckman Coulter Inc., Brea California, United States)

### 4.4. Fluorescence Lifetime Imaging of Cells

MDA-MB-231 cell lines were grown in Dulbecco’s modified Eagle medium (DMEM). Cells were seeded onto 20 mm square glass cover slides into 6-well plates and cultured (2.5 × 10^5^ cells per plate) at 37 °C in a 5% CO_2_ humidified atmosphere with their respective medium. The cells were incubated with 5 μM of *ACB10* in DMEM medium without phenol red for 1, 4, 24, and 48 h. After incubation, the cells were washed three times with PBS.

Fluorescence lifetime images were recorded with a MicroTime 200 microscope (PicoQuant) equipped with a TCSPC card and two TAU-SPAD-100 avalanche photodiode detectors. A 375 nm pulsed diode laser (LDH-D-C-375, PicoQuant) was used as an excitation source, at a 10 MHz repetition rate and a power of ~0.7 µW. The emission was recorded with a long-pass filter (−519/19 LP). The regions of 80 × 80 µm were scanned with 156 nm/pixel spatial resolution and 2 ms of dwell time. FLIM images were processed using SymphoTime64 software (PicoQuant, Berlin, Germany). The lifetime distribution histograms were obtained from FLIM images and were fitted to the Gaussian curve. The FLIM images were smoothed over 200 nm for clarity of presentation. The emission spectra and the histograms were averaged over 3 independent measurements.

### 4.5. Estimation of ADME Parameters

ADME properties were predicted based on 2D structural models, drawn in ChemBioDraw Ultra version 12.0 software (Cambridge Software) and using the SwissADME online tool [35]. The data were analyzed using the Python scipy.stats library. Statistical validity of the established mathematical models was determined by using two correlation coefficients, Pearson and Spearman.

### 4.6. Statistical Analysis

The in vitro experimental data represent the average of three independent experiments, each performed in triplicate, with error bars showing the standard deviation or error of the triplicates. To determine significant statistical differences, we used the student’s *t*-test or Mann–Whitney. *p*-values lower than 0.05 were considered statistically significant, while *p*-values above 0.05 were considered non-significant: * *p* ≤ 0.05; ** *p* ≤ 0.01; and *** *p* ≤ 0.001. Therefore, 95% was set as the level of significance. The analysis was carried out using the software GraphPad Prism 8.0. The results are presented as means ± SD or SEM of at least three independent experiments, each of them performed in triplicate.

## 5. Conclusions

In conclusion, the structure of *ACB3* might serve as the basis for the design of more active guanidine-based antitumoral compounds and pave the way for further chemical optimization and preclinical characterization. In line with this, the readily accessible synthesis of such compounds via guanylation reactions makes *ACB3* a good starting point for improving the pharmacological profile of phenyl-guanidine derivatives for further in vivo evaluation of guanidines as potential chemotherapeutic agents against cancer.

## Figures and Tables

**Figure 1 ijms-24-13820-f001:**
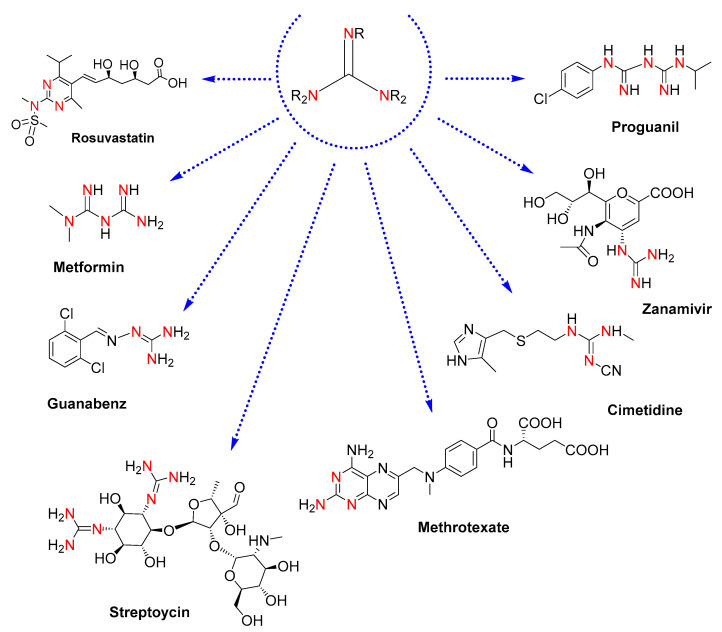
Molecular structures of drugs containing guanidine moieties. The colored nitrogen atoms highlight the guanidine cores present in each molecule.

**Figure 2 ijms-24-13820-f002:**
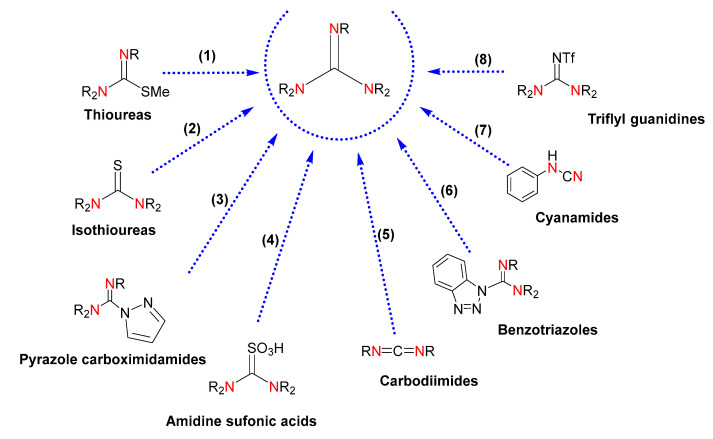
Guanidines obtained by chemical transformation from (1) thioureas, (2) isothioureas, (3) pyrazole carboximidamides, (4) amidine sulfonic acids, (5) carbodiimides, (6) benzotriazoles, (7) cyanamides, and (8) triflyl guanidines. The labeled nitrogen atoms highlight the parts of the starting reagents that will give rise to a common guanidine core in the product, also colored.

**Figure 3 ijms-24-13820-f003:**
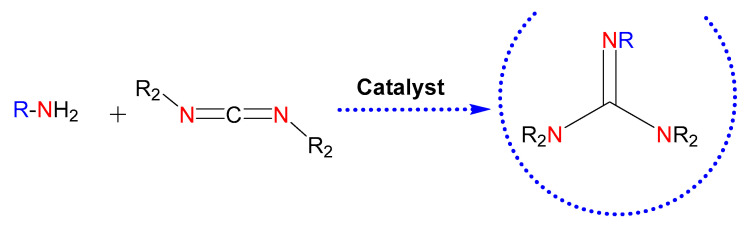
Guanidines obtained by catalysis. The nitrogen atoms in red highlight the parts of the starting reagents that will give rise to the guanidine core in the product, also colored. Those colored in blue refer to possible substituents of the starting amine.

**Figure 4 ijms-24-13820-f004:**
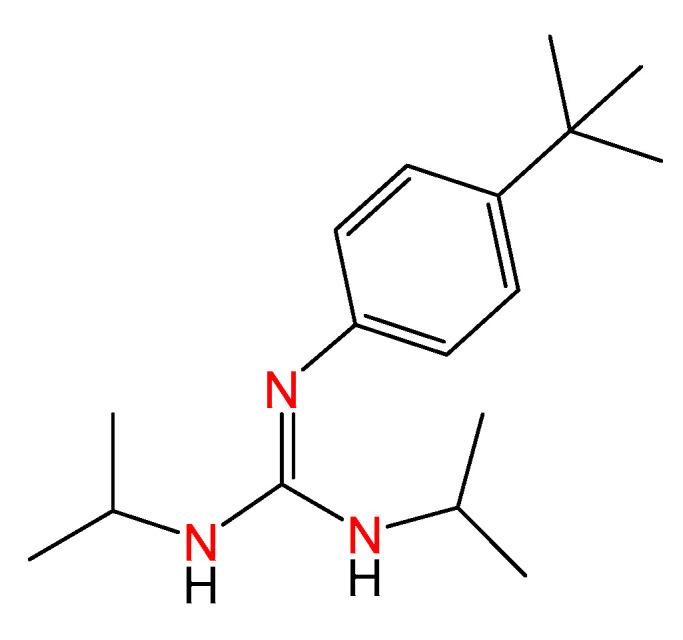
Molecular structure of the previous lead compound *ACB0*. The colored nitrogen atoms highlight the guanidine core.

**Figure 5 ijms-24-13820-f005:**
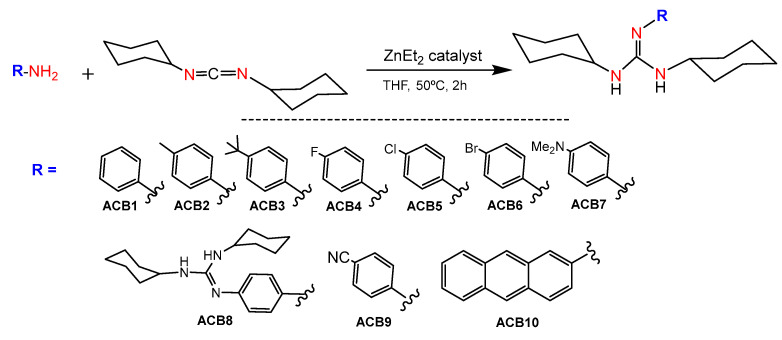
Synthesis of *ACB1-ACB10*.

**Figure 6 ijms-24-13820-f006:**
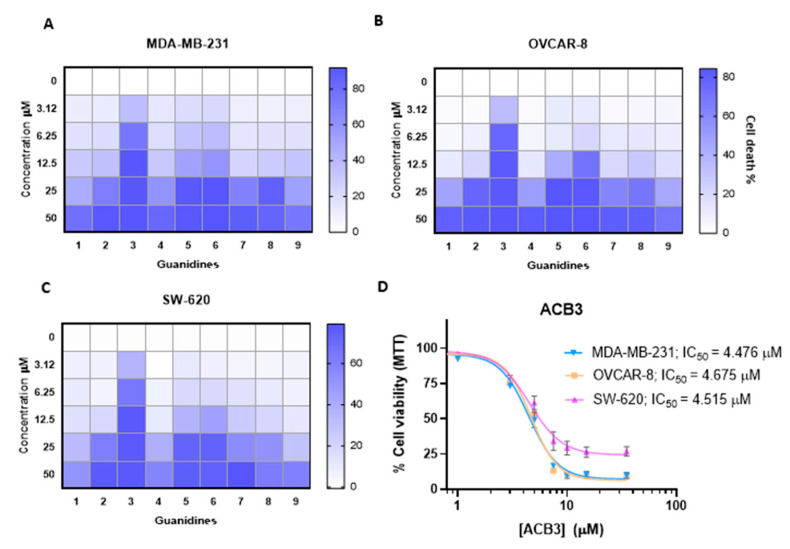
Antitumor activity evaluation of guanidines in three different cancer cell lines. (**A**–**C**) The HeatMap figure visually depicting the antitumor activity of guanidines evaluated by MTT assay for MDA-MB-231 (**A**), OVCAR-8 (**B**), and SW-620 (**C**) cell lines. Cells are treated with the nine tested guanidines (*ACB1-ACB9*) at the five different doses indicated (72 h). The increased color intensity in the grids of the HeatMaps is directly related to a higher % cell death, as indicated in the bar at the top right in the legend. Therefore, as can be seen, *ACB3* guanidine has the highest antitumor activity in all three cancer cell lines and is thus the one we chose for in-depth study. The mean of three experiments is plotted. (**D**) The antiproliferative effect induced by different concentrations of *ACB3* in the indicated cell lines at 72 h was tested by MTT assay. The inhibitory concentration 50 (IC_50_) for each cell line is shown. Calculation of the concentrations IC_50_ based on a nonlinear regression curve fit (log(inhibitor) vs. response-variable slope). The mean of at least three independent experiments (each in triplicate) is plotted together with the standard error (SEM).

**Figure 7 ijms-24-13820-f007:**
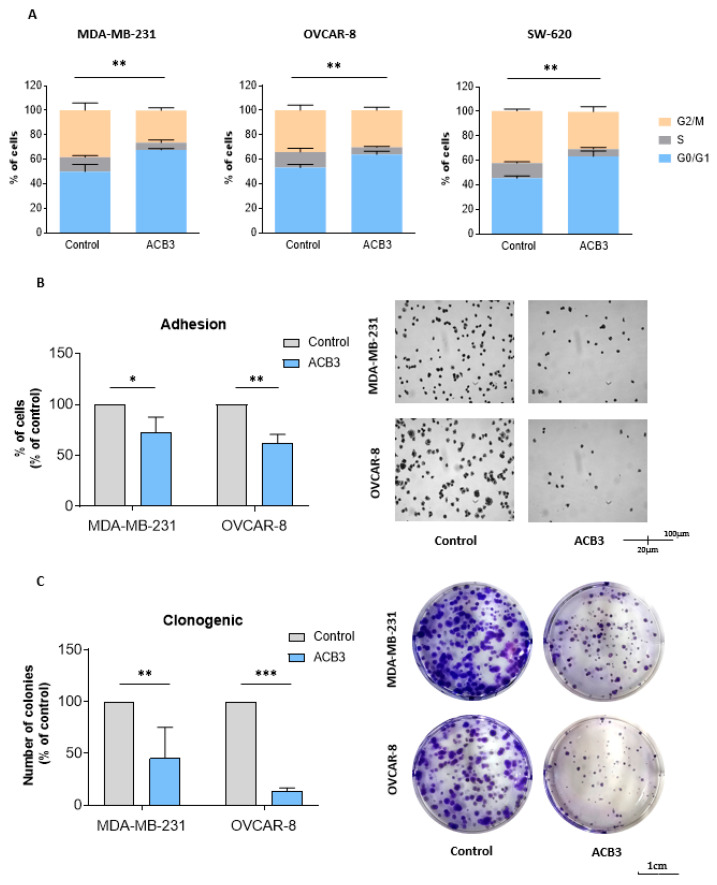
Effect of *ACB3* guanidine on cell cycle, adhesion, and colony formation. (**A**) Assessment of the influence of *ACB3* guanidine on the cell cycle by flow cytometry of MDA-MB-231 (left), OVCAR-8 (middle), and SW-620 (right) cancer cell lines. Bar graphs show the percentage of cells in G0/G1, S, or G2/M cell cycle phases of treated (10 μM, for 24 h) and non-treated cells to compare them. (**B**) To evaluate the effect of the guanidine *ACB3* on cell adhesion, MDA-MB-231 and OVCAR-8 cell lines were treated with this molecule (10 μM) for 24 h. Bar graphs (left) show the percentage of treated cells that have adhered to the plate relative to the control (untreated cells). Microscope images (right) show representative examples of the adhesion experiment. (**C**) These images show colony formation ability after 24 h exposure to *ACB3* (10 μM) in MDA-MB-231 and OVCAR-8 cell lines. The bar diagram (left) represents the percentage of colony numbers of guanidine-exposed cells concerning the control. The images taken by the microscope (right) depict the clonogenic experiment in a representative way. All three replicates for each sample are represented as mean +/− standard deviation (SD). * *p* <0.05; ** *p* < 0.01; *** *p* < 0.001.

**Figure 8 ijms-24-13820-f008:**
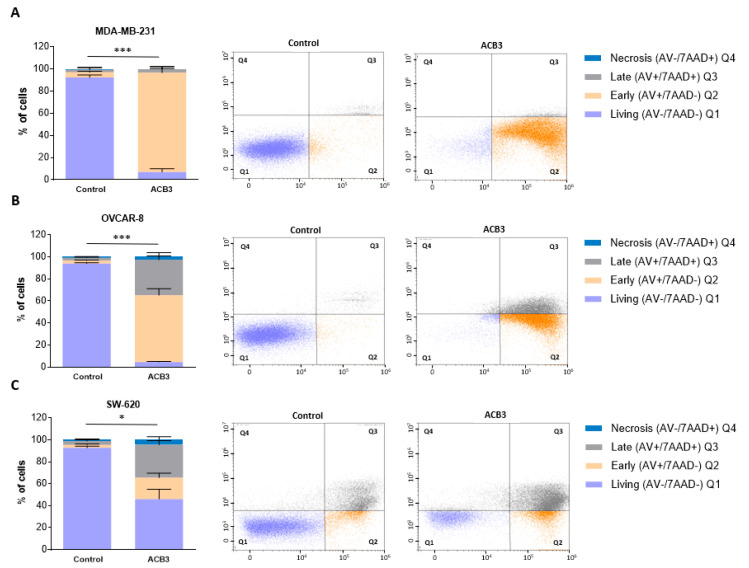
Cell death and apoptosis evaluation of MDA-MB-231 (**A**), OVCAR-8 (**B**), and SW-620 (**C**) cell lines under *ACB3* treatment. (**A**–**C**) The figure displays bar diagrams (left) showing the percentage of cells in four different stages of cell death after 72 h of *ACB3* treatment (10 μM), compared with the untreated cells of the control. All three replicates for each sample are represented as mean +/− standard deviation (SD). Living cells in purple (AV−/7AAD−), early apoptosis in orange (AV+/7AAD−), late apoptosis in gray (AV+/7AAD+) and necrosis in blue (AV−/7AAD+) are the differentiated stages of apoptosis displayed in the picture. Flow cytometry dot-plots (right) provide a more visually representative diagram of the stages of apoptosis for each cell line (Q1 (living), Q2 (early apoptosis), Q3 (late apoptosis), Q4 (necrosis)). * *p* <0.05; *** *p* < 0.001.

**Figure 9 ijms-24-13820-f009:**
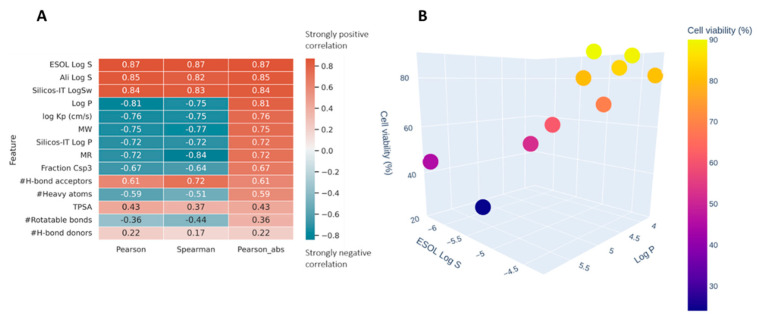
(**A**) Spearman and Pearson correlations between ADME properties of ACB1-ACB9 and cell viability obtained using the in silico method; sorted by the absolute value of Pearson. (**B**) The relationship between *Log P* (*X*-axis), ESOL Log S (*Y*-axis), and the measured cell viability (color and *Z*-axis).

**Figure 10 ijms-24-13820-f010:**
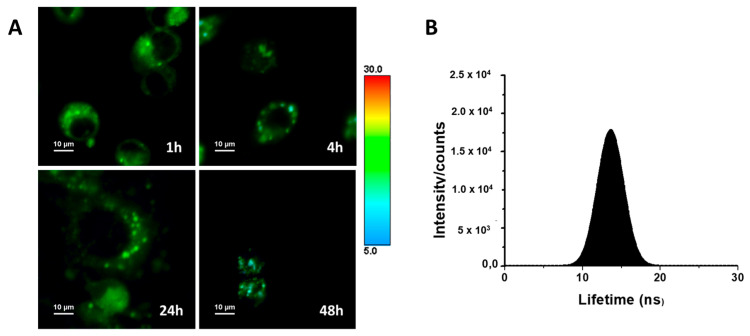
(**A**) FLIM images of the studied cell lines MDA-MB231 with *ACB10* at different incubation times: 1, 4, 24, and 48 h. (**B**) Overall histogram of the average emission lifetimes.

## Data Availability

Not applicable.

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
