# Peer review of "Guanylation Reactions for the Rational Design of Cancer Therapeutic Agents"

_ijms, 2023, doi:10.3390/ijms241813820_

Round 1

Reviewer 1 Report

The manuscript of Almudena del Campo-Balgurias and co-workers entitled:” Guanylation reactions for the rational design of cancer therapeutic agents” reports the development of 9 guanidine containing small molecules, synthetised by a previously reported procedure (by the same group), and their biological evaluations on 3 different cancer cell lines.

The manuscript is divided into five sections: the introduction helps the reader to have the main background related to the described topic and the main information about guanidine core and its relevance inside the pharmaceutical field. However, doing a list of drugs containing the guanidine core and don’t put a scheme of the reactions including challenges and difficulties on this specific moiety is something missing. This section is, in my humble opinion the one that is less well written, tenses like “Cancer is a devastating disease.” Needs to be adapted more to the context and explained.

Materials and Methods is well written reporting all the data needed to repeat the experiments, however, if I have understood well compounds ACB0, ACB6, ACB8 and ACB9 were previously reported, so there are only 6 new compounds.

The Results section is divided in other 6 sub-section, the synthesis of guanidine-based agents is poor repeating mostly the procedure reported. The scope expansion is minimal, and the methods utilised for the characterization of new small molecules are something related to supplementary information more than the main text. Additionally, the figure 3 reports a scheme where a blue “R” is used as general aryl motif but in A and B part of the scheme totally disappear, and the cyclohexyl moiety is abbreviated incorrectly with iPr instead of Cy, to my opinion this scheme is incorrect and misleading. The antiproliferative activity of guanidines ACB0-ACB9 in cancer cell lines describe the evaluation of the synthetised compounds showing how ACB3 seems superior compared the others. I can understand that cell line SW-620 can’t adhere to the petry dish but my question is would have been possible to change cell line? In this way the comparison on the performed test would have been more relevant and extent. Anyway, ACB3 induce colonies reductions into MDA-MB-231 and OVCAR-8 cell lines. The fluorescent study is well performed and highlights something new that could have been more investigated, even if I ask myself if these molecules can stack together and, in this way, impaired the cell machinery?

The discussion and conclusion are well written and underline the main findings of the manuscript.

I believe that the authors did a good work, but to my personal opinion is not novel enough for being published into IJMS, I encourage the authors to extent the scope of ACB derivatives and to focus on fluorescence studies more than in silico to obtain more consistent results.

Written previously

Author Response

Reviewer 1

Comments and Suggestions for Authors

The manuscript of Almudena del Campo-Balgurias and co-workers entitled:” Guanylation reactions for the rational design of cancer therapeutic agents” reports the development of 9 guanidine containing small molecules, synthetised by a previously reported procedure (by the same group), and their biological evaluations on 3 different cancer cell lines.

The manuscript is divided into five sections: the introduction helps the reader to have the main background related to the described topic and the main information about guanidine core and its relevance inside the pharmaceutical field. However, doing a list of drugs containing the guanidine core and don’t put a scheme of the reactions including challenges and difficulties on this specific moiety is something missing. This section is, in my humble opinion the one that is less well written, tenses like “Cancer is a devastating disease.” Needs to be adapted more to the context and explained.

Authors response. We appreciate the reviewer’s positive view of our study and thank the suggestions to improve the quality of the manuscript. Following suggestions, the introduction in a revised manuscript contains new schemes to discuss difficulties in obtaining this specific moiety by traditional organic chemistry. Likewise, comments about cancer treatment have been adapted to the current context.

Materials and Methods is well written reporting all the data needed to repeat the experiments, however, if I have understood well compounds ACB0, ACB6, ACB8 and ACB9 were previously reported, so there are only 6 new compounds.

Authors response. Only the full characterization of novel compounds is included in the main text. The characterization of those that were reported elsewhere by different methodologies is comprised in the supporting information section.

The Results section is divided into other 6 sub-section, the synthesis of guanidine-based agents is poor repeating mostly the procedure reported. The scope expansion is minimal, and the methods utilised for the characterization of new small molecules are something related to supplementary information more than the main text.

Authors response. Guanylation reactions were exploited as a tool for the straightforward generation of a set of guanidine derivatives for the screening of their antitumoral activity Therefore, the methodology to obtain them seems to be repetitive. However, the optimization of the catalyzed synthesis, the characterization of compounds and purification devoted a large amount of time. Most of the guanidines reported in this manuscript have been obtained before by different methodologies, therefore the characterization of the small molecules must go to the supporting information. Following reviewer suggestions, more emphasis was placed on the synthesis of these compounds using catalysis in the results section of the revised manuscript.

Regarding the scope expansion of our synthesis, we would like to explain that in the first step of our work, we synthesized a set of 21 guanidine derivatives using guanylation reactions. Preliminary results on the antitumor activity evaluation of such guanidines in three different cancer cell lines let us rule out those guanidines with isopropyl moieties in their core as potential antitumor agents, as you can see in the figure attached, and a new set of guanidine molecules containing the cyclohexyl moieties were proposed to be synthesized and carried out their antitumour activity evaluation. Such molecules are the ones reported in our manuscript. We decided that the full evaluation of the antitumor activity of more than 20 molecules (those with the isopropyl moieties) with such poor activity was not worth it and very costly to accomplish.

Additionally, the figure 3 reports a scheme where a blue “R” is used as general aryl motif but in A and B part of the scheme totally disappear, and the cyclohexyl moiety is abbreviated incorrectly with iPr instead of Cy, to my opinion this scheme is incorrect and misleading.

Authors response. The figure has been redone in the revised manuscript and following reviewer suggestion to not mislead readers.

The antiproliferative activity of guanidines ACB0-ACB9 in cancer cell lines describe the evaluation of the synthetised compounds showing how ACB3 seems superior compared the others. I can understand that cell line SW-620 can’t adhere to the petry dish but my question is would have been possible to change cell line? In this way the comparison on the performed test would have been more relevant and extent. Anyway, ACB3 induce colonies reductions into MDA-MB-231 and OVCAR-8 cell lines.

Authors response. Although we agree with the fact that an additional cell line would have strengthened the data, all experiments have been performed in three different validated cell lines. We consider that additional cell lines used only to confirm the effect in cell adhesion and colony formation assays would have not changed or modified the biological finding reported.

The fluorescent study is well performed and highlights something new that could have been more investigated, even if I ask myself if these molecules can stack together and, in this way, impair the cell machinery?

Authors response. We sincerely welcome the reviewer's comment. Regarding fluorescence studies, there is much controversy in discerning the mechanism of action of guanidine derivatives. In the bibliography, you can find works with molecules with targets as diverse as the nucleus, cytoplasm, mitochondria, membranes, reticulum, etc... And action mechanisms at different levels such as proteins or DNA (Guanidines as Reagents and Catalysts I, Springer Cham, 2017; ACS Med. Chem. Lett. 2022, 13, 3, 463–469; J. Med. Chem. 2013, 56, 3, 700–711; Med. Chem. Commun., 2019, 10, 26-40; Bioorganic Chemistry 138 (2023) 106600, etc…) That is why here we proposed fluorescent labelling to obtain more information about it. In this sense, FLIM studies allow us to conclude that our guanidine derivatives do not undergo chemical modifications within the cell and that they are not capable of reaching the cell nucleus. Unfortunately, due to the resolution of FLIM technique, we cannot provide more detailed information on the mechanism of action in the cytoplasm. Perhaps with high-resolution imaging techniques, it could be done, but we currently do not have access to them. Regarding your question about possible stacking, because the guanidine group is a superbase and these derivatives will normally be found protonated, the traditional π-π stacks between two aromatic monomers might not be favoured due to the repulsion between them. However, in our laboratory, we have been studying systems based on guanidine moiety where we have observed stacking assisted by bridging water molecules that could give rise to self-assembly between monomers that may impair the cell machinery as the reviewer suggests (Chem. Commun., 2020,56, 4102-4105; ACS Sens. 2021, 6, 9, 3224–3233).

The discussion and conclusion are well written and underline the main findings of the manuscript. I believe that the authors did a good work, but to my personal opinion is not novel enough to be published into IJMS, I encourage the authors to extent the scope of ACB derivatives and to focus on fluorescence studies more than in silico to obtain more consistent results.

Authors response. Unfortunately, we do not agree with this comment at all. Chemotherapeutics have been a much-demanded entity in medicinal chemistry for ages due to their high cytotoxicity. Trial-and-error or learning by doing has governed the research in this field of chemistry, and its principal still-active task is the search for structure-performance relationships. In this regard, guanidine derivatives have been explored as chemotherapeutics for the treatment of many pathologies. However, their synthesis is not efficient and therefore costly. The main aim of this work was to report guanylation reactions as a tool for the straightforward generation of guanidine-core molecules as antitumoral agents. For this purpose, an easy-tunable scaffold was selected for further generation of guanidine-core chemotherapeutics, as a proof of concept. The work reports the use of guanylation reactions catalysed by friendly catalysts to give rise to a family of guanidine derivatives with potential as therapeutic agents in the treatment of cancer. From this work, we identified ACB3 as the lead compound for which biochemical studies might serve as the basis for the design of more active guanine-based antitumoral compounds and pave the way for further chemical optimization and preclinical characterization. To propose a SAR, we synthesized a novel fluorescent guanidine and tracked its cellular uptake using FLIM. Similarly, ADME parameters were obtained by collaborating with the private company Cancerappy and find out the close relationship between lipophilia and activity which could justify the introduction of cyclohexyl groups in the guanidine structure, as we hypothesized before after reporting our first work concerning the use of guanidine in cancer, RSC Adv. 2016, 6, 8267, in which we tried to report novel therapeutic agents for the treatment of glioblastoma. Finally, we would like to say that the lead compound is three times more “potent” than cisplatin, a benchmark drug for the treatment of solid tumours. 

Given the above, we are convinced that this work compiles results enough to make room for a full paper in a prestigious journal such as IJMS. This contribution could be of interest to chemists who applied their knowledge in the design of new antitumoral agents, and physicians who look for structure-activity relationships.

We have modified the manuscript considering all the reviewers’ comments. We frankly hope that the reviewer finds the manuscript improved.

Reviewer 2 Report

This is an interesting, well written, and generally well conceived study.  Most of my concerns are minor, and relate to acknowledging and accounting for the fact that various compounds in the guanidine family have in vivo chemical reactivity that can alter their pharmacological and toxicological profiles.  Specific concerns include the following:

Many guanidine analogs are chemically reactive in vivo, reacting with electrophiles, activating Bronsted acids, etc.  What considerations should be made to prevent off-target effects that many arise from non-pharmacologic chemical reactivity?  Note, of course, that in some applications (especially antimicrobial and anticancer targeting) may partially exploit such reactivity, so a degree of balance may be required (i.e., not so reactive as to modulate host physiology, but enough to interfere with pathogens).

The guanidine miety in some chemicals such as metformin tend, by virtue of their reactive tendencies, to induce hypoglycemic fluxes that can rise to the level of seizure.  What fraction of guanidine analogs might pose glycemic modulation risks?  Can this be predicted?

Given the reactive nature of many quanidines, which can entail complexation with diverse in situ electrophiles, the compound that progresses into plasma may be chemically different from the one synthetically formulated.  Do real experimental ADME measurements agree with in silico predictions?  It would be helpful to include a few controls to address this question.

The manuscript is well written.  I did not pay close attention to minor typographical issues, though there are almost certainly a few.  One that I did catch was fairly subtle:

that require toxic, poor available and costly reagents

  >> that require toxic, poorly available and costly reagents

Author Response

Reviewer 2

This is an interesting, well written, and generally well conceived study.  Most of my concerns are minor, and relate to acknowledging and accounting for the fact that various compounds in the guanidine family have in vivo chemical reactivity that can alter their pharmacological and toxicological profiles. 

Authors response. We thank the reviewer for devoting time to read, thoroughly evaluate and provide constructive criticism to our manuscript.

Specific concerns include the following:

Many guanidine analogs are chemically reactive in vivo, reacting with electrophiles, activating Bronsted acids, etc.  What considerations should be made to prevent off-target effects that many arise from non-pharmacologic chemical reactivity?  Note, of course, that in some applications (especially antimicrobial and anticancer targeting) may partially exploit such reactivity, so a degree of balance may be required (i.e., not so reactive as to modulate host physiology, but enough to interfere with pathogens).

Authors response. We appreciate your interest in our work and all your questions arise regards with what we are already working on in our lab. The design of guanidines faces different limitations particularly when translating into clinical scenarios, where no target specificity exists, and a narrow therapeutic index could be anticipated. To prevent off-target effects of this kind of molecule, we have several strategies. First, the improvement in the pharmacology profile of our entities by the use of synthesis. Novel options to improve this process are associated with the current capabilities of integrating massive information using bioinformatic tools and big data-targeted therapies against selected targets. Second, the potential use of drug delivery systems to improve the activity and safety. The controlled delivery of high concentrations of guanidines by EPR effect or active targeting (guided NPs) might reduce their effects on non-transformed tissue. Third, the use of the feasibility of the guanylation reactions to generate guanidine cores to directly conjugate antibodies and obtain ADCs.

The guanidine miety in some chemicals such as metformin tend, by virtue of their reactive tendencies, to induce hypoglycemic fluxes that can rise to the level of seizure.  What fraction of guanidine analogs might pose glycemic modulation risks?  Can this be predicted?

Authors response. Although we acknowledge that guanidine analogues could induce modifications of the glucose levels, we do not consider that this will be a real limitation for their clinical development. Metformin is approved for type 2 diabetes and rarely induces hypoglycemia which produces a patient risk. Clinical studies with metformin at very high doses have been performed for the treatment of cancer. In addition, other compounds with a much more potential hypoglycemic risk have been evaluated clinically like those antibodies acting against the insulin-like growth factor. In parallel, agents targeting the PI3K/mTOR pathway also produce de-regulation of the glucose level in terms of hypoglycemia and have been approved in several indications.

In summary, we acknowledge the fact suggested by the reviewer, but taking into consideration the adverse events of metformin, the high therapeutic index, and the experience with other compounds, we do not consider this as a limitation.

Given the reactive nature of many quanidines, which can entail complexation with diverse in situ electrophiles, the compound that progresses into plasma may be chemically different from the one synthetically formulated.  Do real experimental ADME measurements agree with in silico predictions?  It would be helpful to include a few controls to address this question.

Authors response. We completely agree with the reviewer regarding the significance of validating our in-silico predictions. In fact, we are already looking into it. First, lipophilicity for what we are calculating log P and log D by LC/MS/MS measurement of compounds. Second, solubility by UV spectrophotometry measurements. Third, plasma stability by LC/MS/MS detection. Fourth, permeability using a lipid bilayer established on a membrane filter. Unfortunately, these experiments are time consuming, being far away from the 10-days time left by the journal to answer the comments. Hopefully, these results will be included into following works.

Comments on the Quality of English Language

The manuscript is well written.  I did not pay close attention to minor typographical issues, though there are almost certainly a few.  One that I did catch was fairly subtle:

that require toxic, poor available and costly reagents

  >> that require toxic, poorly available and costly reagents

Authors response. The revised manuscript was proofread again before resubmitting.

Round 2

Reviewer 1 Report

The original manuscript has been improved by authors following the suggestions, I have really appreciated it.
I personally think that now it looks better and clearer than before, it can be accepted by IJMS.